# Optimal Strategies to Select Warfarin Dose for Thai Patients with Atrial Fibrillation

**DOI:** 10.3390/jcm13092675

**Published:** 2024-05-02

**Authors:** Anunya Ujjin, Wanwarang Wongcharoen, Arisara Suwanagool, Chatree Chai-Adisaksopha

**Affiliations:** 1Division of Medicine, Neurological Institute of Thailand, Bangkok 10400, Thailand; 2Division of Cardiology, Department of Internal Medicine, Faculty of Medicine, Chiang Mai University, Chiang Mai 52000, Thailand; 3Division of Cardiology, Department of Internal Medicine, Faculty of Medicine, Siriraj Hospital, Bangkok 10700, Thailand; 4Division of Hematology, Department of Internal Medicine, Faculty of Medicine, Chiang Mai University, Chiang Mai 52000, Thailand

**Keywords:** warfarin, atrial fibrillation, warfarin dose

## Abstract

**Background**: Warfarin has been the mainstay treatment for the prevention of stroke and systemic thromboembolism in patients with atrial fibrillation (AF). The optimal starting dose of warfarin remains unclear. **Objective**: To investigate the most optimal dosing strategies for warfarin starting dose in Thai patients with AF. **Material and Methods**: We enrolled consecutive AF patients who were starting on warfarin and resulting in a stable INR of 2.0–3.0 at two consecutive time points. We measured the dose of warfarin at which INR achieved the target range. The optimal dosage was defined as the difference from the actual dose within 20%. We compared strategies of warfarin dosing, including warfarin dosing formula, 2.5 mg, 3 mg and 5 mg doses. The primary endpoints were the proportions of patients in optimal, underdosing, and overdosing categories. **Results**: Among 1207 patients visiting the Outpatient Clinic between October 2011 and September 2021, 531 patients were identified with AF and INR in the therapeutic range of 2.0–3.0 on at least two consecutive visits. The mean age of participants was 68 ± 11 years, and men accounted for 44.4% of the population. The warfarin dosing formula resulted in optimal dosing in 37% and overdosing in 24% of cases, whereas the 2.5 mg, 3 mg and 5 mg doses resulted in optimal dosing in 36%, 39%, and 11%, and overdosing in 33%, 44% and 88% of patients, respectively (*p* < 0.01). **Conclusions**: In Thai patients with AF, the optimal warfarin starting dose may be 2.5 mg, 3 mg or a simplified warfarin dosing formula, whereas the 5 mg dose should be avoided due to the high risk of overdosing.

## 1. Introduction

Although direct oral anticoagulants (DOACs) have become widely used, the development of warfarin usage remains [1]. A recent population-based study reported that approximately 18% of patients with non-valvular atrial fibrillation (AF) continued using warfarin as an anticoagulant [2].

Stroke is the most devastating thromboembolic event in AF. Moreover, approximately 36% of all strokes in individuals aged 80 to 90 years are attributed to AF. Not all AF patients are at high risk for stroke. The recognized clinical markers predicting increased risk for stroke in AF are a prior stroke or transient ischemic attack, hypertension, diabetes mellitus, congestive heart failure, and age older than 75 years [2,3]. Other less validated stroke risk factors include coronary artery disease, thyrotoxicosis, female sex, LV dysfunction, and age older than 65 years [4,5]. Mitral stenosis is well known to be associated with a high risk for stroke in AF patients.

The European Society of Cardiology (ESC) 2020 management guidelines recommend that AF patients identified as low-risk (CHA_2_DS_2_-VASc 0 (males), or score of 1 (females)) do not need any stroke prevention treatment due to consistently low ischemic stroke or mortality rates (<1%/year). In the presence of a > 1 non-sex stroke risk factor, women with AF have significantly higher stroke risk than men, and should initiate anticoagulant therapy [6].

Warfarin, a conventional anticoagulant drug, has a complex range of benefits and difficulties related to its therapeutic use. The normalization of D-dimer levels was observed among atrial fibrillation patients receiving warfarin, which is consistent with the antithrombotic effect of warfarin [7]. Although it is acknowledged for its effectiveness in reducing thromboembolic events, especially strokes, the therapeutic administration of this treatment requires careful attention due to its limited therapeutic range. The regular monitoring of international normalized ratio (INR) levels and rigorous compliance with dietary restrictions, particularly regarding vitamin K consumption, are necessary. Although it is cost-effective and can reverse acute bleeding, the practical problems of constant monitoring and possible dosage variability make it difficult to implement. Current research endeavors aim to elucidate the relative efficacy and safety profiles of warfarin compared to DOACs. A recent real-world study conducted by Kundnani et al. reported that non-valvular AF patients who took apixaban had a lower rate of thromboembolic events than those who took acenocoumarol [8]. However, warfarin is indicated in some circumstances, for example, advanced stages of chronic kidney disease or valvular heart disease. A recent study led by Kreutz reported only 5.8% of AF patients with CKD did not receive any type of anticoagulant [9].

Despite its affordability, warfarin’s clinical utility is tempered by the substantial incidence of bleeding complications, particularly intracranial hemorrhage, thus challenging its overall effectiveness in clinical practice [10].

Giving an optimal starting dose of warfarin may prevent adverse outcomes. However, there is no such standard recommendation regarding the selection of warfarin starting dose. A previous small study reported that a 3 mg initiating dose of warfarin appeared to be safe in Asian patients [11]. The independent predictors of high INR variability are symptomatic heart failure, older age (>75 years old), and severe renal dysfunction [12].

Pongbangli et al. introduced a simplified warfarin dosing formula aimed at guiding the initiating dose in Thai patients who required warfarin therapy [13]. This study utilized age, body weight, and history of congestive heart failure and/or stroke in the simplified warfarin predicting formula. The authors reported that the warfarin dosing formula resulted in a reasonable number of patients falling into optimal dosing categories.

The objective of this present study was to compare the optimal strategies to select the initial warfarin dose.

## 2. Materials and Methods

The present study was a single-center, retrospective cohort study of patients who were enrolled between 1 October 2011 and 30 September 2021 at a referral hospital. The study included patients with AF, categorized into two main types: non-valvular AF and rheumatic mitral stenosis-associated AF. Non-valvular AF refers to AF occurring in the absence of significant valvular disease, such as mitral valve stenosis or mechanical heart valves. This subgroup comprised patients with AF not attributable to valvular abnormalities and accounted for the majority of cases in the study population. The inclusion criteria of the study were as follows: (i) patients had AF, regardless of whether they presented with the first episode or recurrent episodes of AF, (ii) patients had undergone continuous warfarin therapy with the target INR of 2.0 to 3.0 at the Outpatient Clinic for more than three months, regardless of their initial OAC treatment status, including the potential use of LMWH, and (iii) patients had achieved INR targets for at least two consecutive follow-up visits. Sequential exclusion criteria were applied to mitigate confounding factors and ensure the homogeneity of the study cohort. These criteria included the exclusion of patients with mechanical heart valves, post-mitral valve surgery, venous thromboembolism (VTE), foreign patients, and those failing to achieve therapeutic INR levels (2.0 to 3.0) for at least two consecutive follow-up visits, as shown in Figure 1. The electronic medical records of patients were reviewed. Baseline characteristics, including age, body weight, body height, comorbidities, concurrent medications, actual warfarin dose, and INR levels, were collected. The study protocol was approved by the institutional Medical Ethics Committee project approval number S002b/65ExPD.

In our country, warfarin is typically marketed in tablet form, with common dosages including 2 mg, 3 mg, and 5 mg tablets. When prescribing warfarin, dosages such as 1.5 mg, 2 mg, 2.5 mg, 3 mg, or 5 mg may be required based on factors such as patient characteristics and target INR range. To achieve these dosages, healthcare providers may prescribe a combination of different tablet strengths or tablet splitting to achieve the desired dosage.

Warfarin dosage was prescribed and adjusted by treating physicians based on INR results. We collected the dose of warfarin when INR achieved the target range of 2.0 to 3.0 in two consecutive follow-ups and observed the duration required to achieve the INR target in the patients for whom we have data regarding the initiation of warfarin.

The warfarin dosing formula used in the present study was derived using the algorithm described by Pongbangli N et al. and Sarapakidi et al. [13]. The simplified warfarin dosing formula (mg/day) was = 3.2 − (0.03 × age (years)) + (0.02 × body weight (kg)) (10% dose reduction if the presence of heart failure or stroke)—warfarin dosing Formula (1) [13,14].

We adapted a simplified formula by omitting a history of heart failure and stroke (mg/day) = 3.2 − (0.03 × age (years)) + (0.02 × body weight (kg)) without using body weight and the presence of heart failure or stroke—Warfarin dosing Formula (2).

The other dose strategies commonly used in practice are the 2.5 mg dose, 3 mg dose, and 5 mg dose.

The actual warfarin dose was defined as the warfarin dose that resulted in INR 2.0 to 3.0 for at least two consecutive follow-ups after the warfarin initiation. The optimal dosage was defined as the difference from the actual dose beginning within 20%.

The simplified formula was developed in Chiang Mai, Thailand, and this study sought external validation in Sakaeo province, Thailand. This two-site validation ensures the formula’s applicability and reliability across different geographic regions and patient populations. Additionally, we compared the actual dose of warfarin administered to patients with the warfarin dose strategies. These strategies included doses of 2.5 mg, 3 mg, and 5 mg; this comparison aimed to assess the appropriateness and effectiveness of these dose strategies in achieving the target INR range of 2.0 to 3.0.

The primary endpoint of the study is to compare the proportion of patients falling into optimal dosing, underdosing, and overdosing when using the simplified warfarin dosing formula, compared with doses of 2.5 mg, 3 mg, and 5 mg. The secondary endpoints included the actual warfarin starting dose, days to achieve target INR, and bleeding outcomes.

## 3. Statistical Analysis

Continuous variables were presented as mean ± standard deviation (SD) or median and interquartile range. Categorical variables were displayed as percentages. Differences between continuous variables were assessed using an unpaired 2-tailed *t*-test for normally distributed continuous variables and the Mann–Whitney test for skewed variables. Proportions were compared using a Chi-square test of Fisher’s exact test when appropriate. The level of agreement between the predicted dose and the actual dose was demonstrated using a modified Blant–Altman plot. All statistical significances were set at a *p*-value less than 0.05 and all statistical analyses were carried out using Stata 16.0.

## 4. Results

### 4.1. Baseline Characteristics

Five hundred and thirty-one patients receiving warfarin who had achieved the target INR of 2.0–3.0 were included in the study. The mean age ± SD was 68 ± 11 years, and men accounted for 44.4% of the population (n = 236). The mean body weight was 61.6 ± 14 kg. Non-valvular AF was presented in 416 (78.34%). The mean CHA_2_DS_2_-VASc Score was 3.65 ± 1.27. No patients were taking carbamazepine, phenytoin, or rifampicin. Amiodarone was prescribed for two patients (0.38%). The baseline characteristics of the studied population are shown in Table 1.

### 4.2. Actual Warfarin Starting Dose

The majority of patients (55.9%) started on warfarin 1–1.5 mg/day, followed by 3 mg (26.3%) and 2–2.5 (17.8%) mg/day, respectively. The median of initiating dose was 1.5 mg/day (range, 1.5–3). The median number of days on which the INR target was achieved was 91 days. When classifying patients into three groups according to the warfarin starting dose, patients who started with 1–1.5 mg/day achieved the target INR at a median time of 101 days (range 45–203) (Table 2). Patients who started warfarin with 2–2.5 mg/day achieved target INR at a median time of 81 days (range 16–184), and those who started warfarin at 3 mg/day or higher achieved target INR at a median time of 77 days (range 34–213). When comparing three starting dose regimens, patients who started with 3 mg/day or higher took the shortest time to achieve target INR when compared to the other two dose regimens (*p* = 0.037).

### 4.3. Actual Stable Warfarin Maintenance Dose

The mean actual warfarin dose was 2.76 ± 1.16 mg/day. The median of the actual warfarin dose was 2.57 mg/day (IQR 2, 3.2). The minimum and maximum dose requirements were 0.71 mg/day and 7.5 mg/day, respectively. The histogram of the actual warfarin maintenance dose of patients included in the study is shown in Appendix A Figure A1.

### 4.4. Performance of Warfarin Predicting Dose Using Formulas and Fixed-Dose Formulas

When using a higher warfarin dose, the mean predicted warfarin dose was 2.35 ± 0.52 mg/day for Formula (1) and 2.2 ± 0.50 mg/day for Formula (2), respectively. The relationship between the predicted dose and the actual dose of warfarin is shown in Appendix A Figure A2. The mean differences in warfarin dose between actual and simplified formulas were 0.57 ± 1.1 mg/day (0.48–0.66) for Formula (1) and 0.42 ± 1.07 mg/day (0.32–0.51) for Formula (2). The disagreement was more apparent in patients requiring a higher dose of warfarin. Figure 2 demonstrates the comparative performance among warfarin dosing strategies. Warfarin dosing Formulae (1) and (2) resulted in appropriate dosing in 36.72% and 37.09% of patients, respectively. Overdosing was observed in 18.46% for Formula (1) and 24.29% for Formula (2). Underdosing was observed in 44.82% for Formula (1) and 38.60% for Formula (2). For the fixed dosing strategies, 2.5 mg, 3 mg, and 5 mg doses resulted in appropriate dosing in 36.35%, 38.79%, and 10.92%, respectively. Overdosing was observed in 33.33% for the 2.5 mg dose, 43.88% for the 3 mg dose and 87.59% for the 5 mg dose. Underdosing was observed in 30.32% for the 2.5 mg dose, 17.33% for the 3 mg dose and 1.13% for the 5 mg dose.

### 4.5. Bleeding Outcomes

Major bleeding was found in 7.75% of patients. Intracranial hemorrhage was found in 4.54% and gastrointestinal bleeding was found in 3.78%. We compared the major bleeding, intracranial bleeding, and GI bleeding across three groups of warfarin dosage, and there was no statistical significance (Appendix A Table A1).

## 5. Discussion

Despite DOACs showing a better efficacy and safety profile compared with warfarin in patients with AF and a history of bleeding, warfarin is commonly used in clinical practice for the prevention of stroke and systemic thromboembolism in patients with AF in developing countries. The bleeding complications of warfarin are mostly due to the narrow therapeutic index of warfarin and inter- and intraindividual variability in the dose–response of warfarin [15].

A previous study observed temporary and permanent excessive warfarin anticoagulation risk factors; the strongest were excessive alcohol consumption in 9.6% of patients (OR 24.4, 95% CI 9.9–50.4, *p* < 0.0001) and reduced renal function (OR 15.2, 95% CI 5.67–40.7, *p* < 0.0001). Recent use of antibiotic or antifungal medication, recent hospitalization or outpatient clinic visit, and the first six months of warfarin use were the most significant temporary risk factors for excessive warfarin anticoagulation [16]. Another previous study showed that patients with left ventricular systolic dysfunction required a lower dose of warfarin [17].

A previous study showed that standardized warfarin initiation nomograms are safe and effective, and patients’ responses to them could be used to predict warfarin requirements without the need for genetic testing. The maintenance warfarin dose can be accurately predicted using individual responses to a standard warfarin initiation nomogram without the need for costly genetic testing [18].

In general, genetic variation affects warfarin response and the dosage requirement to a larger extent than clinical factors [19]. The genetic polymorphisms of CYP2C9 and VKORC1 genotypes on the pharmacokinetics and pharmacodynamics of warfarin play an important role in predicting the maintenance dose of warfarin. A multicenter randomized clinical trial in Chinese adults showed that the outcomes of genotype-guided warfarin dosing were superior to those of clinical standard dosing [20], and using genetic dosing for warfarin initiation can lead to a better outcome [21,22]. The prevalences of the CYP2C9*1/*1 genotype and VKORC1 haplotype AA range from 95% to 99% and 57% to 63%, respectively, in the Thai population [23]. Haplotype A is found more frequently in Asian populations than in other ethnic groups. As a result, patients from an Asian population require lower warfarin doses than other ethnic populations. However, recent studies showed that forecasting models using early responses to standardized fixed-dose warfarin initiation nomograms have been developed with performances similar or superior to pharmacogenetics-based models, and without the need for costly genetic testing [24].

Warfarin initiation nomograms for VTE have suggested the use of a 10 mg or 5 mg loading dose for the initiation of warfarin to achieve an INR of 2.0 to 3.0 on the fifth day of therapy [25]. However, this study showed that giving warfarin at a 5 mg fixed dose may increase overdosing events.

Genetics-guided warfarin dosing remains controversial and costly. Moreover, genetic testing is not practical for the majority of AF patients in most Asian countries outside of clinical trials. The use of a warfarin dosing formula that does not incorporate genetic profiling is appealing in such a population. The application of the calculated warfarin dosing formula given by Pongbangli N et al. demonstrated optimal dosing in 41% and overdosing in 21% of cases, whereas a 3 mg initiating dose resulted in optimal dosing in 39% and overdosing in 43% of patients. In patients with HF and/or stroke, using the formula resulted in overdosing in 23% of cases, whereas a 3 mg initiating dose led to overdosing in 53% of patients [13].

In the present study, we compared the performances of warfarin dosing Formula (1), warfarin dosing Formula (2), a fixed 2.5 mg dose, a fixed 3 mg dose, and a fixed 5 mg dose in predicting the optimal maintenance dose of warfarin. The fixed 5 mg dose yileded the greatest number of overdoses on warfarin in this study. We suggest that a fixed 2.5 mg dose of warfarin may be used in the Thai population with high bleeding risk, whereas a 3 mg fixed dose of warfarin can be used in a patient with no bleeding risk in the context of an outpatient clinic with insufficient time to use the formula strategy. That said, using warfarin dosing formulas appeared to be safer than 3 mg and 2.5 mg initiating doses.

The present study had some limitations. First, the simplified warfarin dosing formula cannot be used in patients who require a target INR of more than 2.0 to 3.0. Second, our study did not include patients with a mechanical heart valve, who may have a different response to warfarin dose. Third, we did not collect the thrombotic events, which might be complicated by warfarin underdosing. Fourth, we did not collect data regarding time in the therapeutic range. Instead, patients with stable INR were defined as those who had achieved INR targets at least two consecutive follow-up visits. Lastly, we excluded 56% of patients in this cohort due to erratic INR. As mentioned earlier, we had to include patients with stable INR so as to precisely predict warfarin dosage. This might consequently affect the generalizability of the findings.

## 6. Conclusions

Our study suggests that in Thai patients with AF, the best warfarin starting dose may be 2.5 mg or 3 mg, or the use of a simplified warfarin dosing formula, whereas the 5 mg dose should be avoided due to the high risk of overdosing. A prospective study evaluating different warfarin starting strategies should be performed.

## Figures and Tables

**Figure 1 jcm-13-02675-f001:**
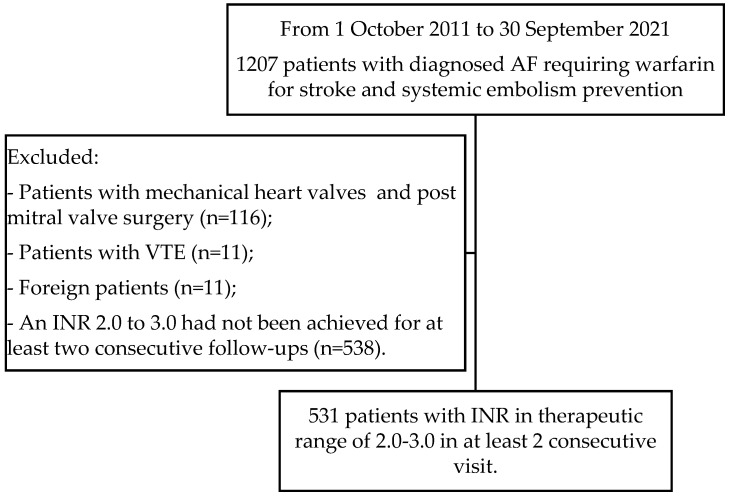
Study population flow chart with inclusion and exclusion criteria used to select 1207 patients.

**Figure 2 jcm-13-02675-f002:**
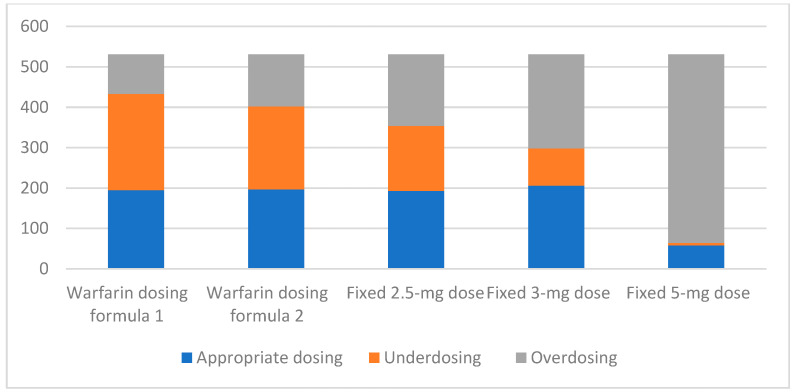
Comparative performance among warfarin dosing Formula (1), Formula (2), fixed 2.5 mg, fixed 3 mg, and fixed 5 mg dose strategies to predicted optimal dose of warfarin. Warfarin dosing Formula (1) = 3.2 − (0.03 × age (years)) + (0.02 × body weight (kg)) (10% dose reduction if the presence of heart failure (HF) and/or stroke). Warfarin dosing Formula (2) = 3.2 − (0.03 × age (years)) + (0.02 × body weight (kg). The appropriate dosage was defined as the difference from the actual dose within 20%.

**Table 1 jcm-13-02675-t001:** Baseline characteristics of studied population.

Baseline Characteristics	Total (n = 531)
Age (years), mean ± SD	68 ± 11
Age > 75 years, n (%)	166 (31.3%)
Male, n (%)	236 (44.4%)
Body weight (kg), mean ± SD	61.6 ± 14
Creatinine (mg/dL), mean ± SD	1.04 ± 0.44
Type of AF	
Non-valvular AF, n (%)	416 (78.3%)
Rheumatic mitral stenosis, n (%)	115 (21.66%)
Diabetes mellitus, n (%)	123 (23.16%)
Hypertension, n (%)	307 (57.92%)
ESRD on RRT, n (%)	2 (0.38%)
Cirrhosis, n (%)	13 (2.45%)
Heart failure, n (%)	191 (36.17%)
History of ischemic stroke, n (%)	186 (35.03%)
Use of amiodarone, n (%)	2 (0.38%)

AF = atrial fibrillation; ESRD = end-stage renal disease; RRT = renal replacement therapy; SD = standard deviation.

**Table 2 jcm-13-02675-t002:** Median of days to achieve INR target in each actual warfarin starting dose.

Actual Warfarin Starting Dose	Days (Median, IQR)
1–1.5 mg	101 (45, 203)
2–2.5 mg	81 (16,184)
3 mg	77 (34, 213)

## Data Availability

The data that support the findings of this study are available on request from the corresponding author, [C.C.-A.]. The data are not publicly available due to restriction of ethics’ policy.

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
