# Peer review of "Optimal Strategies to Select Warfarin Dose for Thai Patients with Atrial Fibrillation"

_jcm, 2024, doi:10.3390/jcm13092675_

Round 1

Reviewer 1 Report

Comments and Suggestions for Authors

Authors performed a retrospective study on 531 AF patients to compare the effectiveness of fixed dose strategy vs simplified dose strategy to reach the INR target in two consecutive samples.

Major

1.      About 56% of cohort was excluded, this is a potential selection bias.

2.      How is the drug warfarin marketed in your country? What dosages are there? How could you prescribe 3 mg, 2.5 mg or 5 mg? I think it is important to specify this in the methods.

3.      Please, in table 1 report data also according to 3 study arm to assess potential baseline confounding characteristics.

4.      Why the simplified formula should be useful compared to previous one? What kind of advantage give to patients?

5.      What is the time to therapeutic range and the time in therapeutic range among patient groups?

6.      What is the definition of non-valvular AF in your study? Please, specify it in the methods.

7.      Authors concluded that 2.5 mg fixed dose may be helpful In patients with major bleeding, however, table 3 showed no difference in bleeding outcomes, the conclusion is not fully supported by data.

8.      Authors reported “In the present study, we compared the performance of a simplified warfarin dosing 228 formula, fixed 2.5-mg dose, fixed 3-mg dose, and fixed 5-mg dose strategy in predicting 229 the optimal maintenance dose of warfarin”, however, data in the study were on 1-1.5, 2.5 or 3 mg, why?

9.      The endpoint of the study is not well describe, I suggest to rephrase the methods section.

Author Response

Response to Reviewer 1 Comments

1. Summary

Thank you very much for taking the time to review this manuscript. Please find the detailed responses below and the corresponding revisions track changes in the re-submitted files

2. Questions for General Evaluation

Reviewer’s Evaluation

Response and Revisions

Does the introduction provide sufficient background and include all relevant references?

Must be improved

We have made revisions to the new version as outlined below

Are all the cited references relevant to the research?

Must be improved

We have made revisions to the new version as outlined below

Is the research design appropriate?

Must be improved

We have made revisions to the new version as outlined below

Are the methods adequately described?

Must be improved

We have made revisions to the new version as outlined below

Are the results clearly presented?

Must be improved

We have made revisions to the new version as outlined below

Are the conclusions supported by the results?

Must be improved

We have made revisions to the new version as outlined below

3. Point-by-point response to Comments and Suggestions for Authors

Comments 1: Authors performed a retrospective study on 531 AF patients to compare the effectiveness of fixed-dose strategy vs simplified dose strategy to reach the INR target in two consecutive samples.

Major

Comment 1: About 56% of cohort was excluded, this is a potential selection bias.

Response 1: Thank you for acknowledging the importance of considering the exclusion of participants based on their INR levels not being within the therapeutic range. The aim of our study is to compare warfarin starting dose strategies. In order to achieve this objective, we have to enroll only patients who have had stable INR.

To address this issue, we have added an additional paragraph stating this limitation in the discussion section.

The present study had some limitations, First, the simplified warfarin dosing formula cannot be used in patients who required a target INR of more than 2.0 to 3.0. Second, our study did not include patients with a mechanical heart valve, which may have a different response to the warfarin dose. Third, we did not collect the thrombotic events, which might be complicated from under warfarin dosing. Lastly, we excluded 56% of patients in this cohort due to erratic INR. As mentioned earlier, we had to include patients with stable INR to precisely predict warfarin dosage. This might consequently affect the generalizability of the findings.

We also have modified the abstract and manuscript to clarify this aspect. Page 1, line 14

Abstract:

Objective: To investigate the most optimal dosing strategies for choosing warfarin starting dose in Thai patients with AF.

Comments 2: How is the drug warfarin marketed in your country? What dosages are there? How could you prescribe 3 mg, 2.5 mg or 5 mg? I think it is important to specify this in the methods.

Response 2: Thank you for highlighting this aspect. In our country, warfarin is typically marketed in tablet form. Common dosages include 2 mg, 3 mg, and 5 mg tablets. When prescribing warfarin, healthcare providers often consider factors such as the patient's age, weight, medical history, and target INR range. To achieve dosages such as 3 mg, 2.5 mg, or 5 mg, healthcare providers may prescribe a combination of different tablet strengths or divided tablets that can achieve the desired dosage. We agree that specifying this information in the methods section would provide clarity regarding the prescription process.

We have modified the method section to emphasize this point as below, Page 3, line 105

   The electronic medical records of patients were reviewed. Baseline characteristics, including age, body weight, body height, comorbidities, concurrent medications, actual warfarin dose, and INR levels were collected. The study protocol was approved by the institutional Medical Ethics Committee project approval number S002b/65ExPD.

   In our country, warfarin is typically marketed in tablet form, with common dosages including 2 mg, 3 mg, and 5 mg tablets. When prescribing warfarin, dosages such as 1.5 mg, 2 mg, 2.5 mg, 3 mg, or 5 mg may be required based on factors such as patient characteristics and target INR range. To achieve these dosages, healthcare providers may prescribe a combination of different tablet strengths or divided tablets that can achieve the desired dosage.

C  Comments 3: Please, in table 1 report data also according to 3 study arm to assess potential baseline confounding characteristics.

Response 3 : Thank you for your comment. Due to the nature of our study design, where patients were not assigned a fixed dose of warfarin but rather had their dosage prescribed and adjusted by treating physicians based on INR results, we only included data from one group of patients in Table 1. This group represents patients who achieved the target INR range of 2.0 to 3.0, and their warfarin dosage was collected at that point. We acknowledge the importance of assessing potential baseline confounding characteristics across different study arms; however, our study design did not allow for such comparisons as there was no allocation to specific treatment arms.

We have modified the method section to clarify this point as below, Page 3, line 111

Warfarin dosage was prescribed and adjusted by treating physicians based on INR results. We collected the dose of warfarin when INR achieved the target range of 2.0 to 3.0 in two consecutive follow-ups and observed the duration required to achieve the INR tar-get in the patients for whom we have data regarding the initiation of warfarin.

The warfarin dosing formula used in the present study was derived using the algorithm described by Pongbangli N et al. and Sarapakidi et al.(13) The simplified warfarin dosing formula (mg/day) was = 3.2 – (0.03 × age (years)) + (0.02 × body weight (kg)) (10% dose reduction if the presence of heart failure or stroke) - Warfarin dosing Formula 1. (13, 14).

    Comments 4: Why the simplified formula should be useful compared to previous one? What kind of advantage give to patients?

Response 4:

       The simplified formula, incorporating body weight, age, heart failure, and stroke history, offers several advantages over the previous method. By considering these factors, it optimizes warfarin dosing, enhancing accuracy and safety in prescription. With a clearer and more tailored dosing regimen, patients are more likely to comply, ultimately leading to better outcomes. Thus, the simplified formula benefits both healthcare providers and patients, promising improved management of warfarin therapy.

        Additionally, the simplified formula was developed at Chiang Mai, Thailand, and this study sought external validation at Sakaeo province, Thailand. This two-site validation ensures the formula's applicability and reliability across different geographic regions and patient populations. Additionally, we compared these formulas with other fixed-dose regimens to assess their performance and utility.

We have modified the method section to emphasize this point as below, Page 3,line 128

The simplified formula was developed at Chiang Mai, Thailand, and this study sought external validation at Sakaeo province, Thailand. This two-site validation ensures the formula's applicability and reliability across different geographic regions and patient populations. Additionally, we compared the actual dose of warfarin administered to patients with the warfarin dose strategies. These strategies included fixed doses of 2.5 mg, 3 mg, and 5 mg, this comparison aimed to assess the appropriateness and effectiveness of these dose strategies in achieving the target INR range of 2.0 to 3.0.

    Comments 5: What is the time to therapeutic range and the time in therapeutic range among patient groups?

    Response 5: The study did not include data on the time to therapeutic range (TTR). We included patients who had been on stable dose of INR, as defined by patients had achieved INR targets for at least two consecutive follow-up visits. We have stated this as the limitation.

    The present study had some limitations, First, the simplified warfarin dosing for-mula cannot be used in patients who required a target INR of more than 2.0 to 3.0. Sec-ond, our study did not include patients with a mechanical heart valve, which may have a different response to the warfarin dose. Third, we did not collect the thrombotic events, which might be complicated from under warfarin dosing. Fourth, we did not collect the data regarding time in therapeutic range. Instead, patient with stable INR was defined as those who had achieved INR targets for at least two consecutive follow-up visits. Lastly, we excluded 56% of patients in this cohort due to erratic INR. As mentioned earlier, we had to include patients with stable INR in order to precisely predict warfarin dosage. This might consequently affect the generalizability of the findings.

    Comments 6: What is the definition of non-valvular AF in your study? Please, specify it in the methods.  

    Response 6: In our study, non-valvular atrial fibrillation (AF) was defined as atrial fibrillation in the absence of rheumatic mitral stenosis, a mechanical or bioprosthetic heart valve, or mitral valve repair. This definition was specified in the Methods section to ensure clarity regarding the population included in the study. Thank you for highlighting the importance of providing this definition.       

We have modified the method section to emphasize this point as below, Page 2, line 85

The present study was a single-center, retrospective cohort study of patients who were enrolled between 1 October 2011 and 30 September 2021 at a referral hospital. The study included patients with atrial fibrillation (AF), categorized into two main types: non-valvular AF and rheumatic mitral stenosis-associated AF. Non-valvular AF refers to AF occurring in the absence of significant valvular disease, such as mitral valve stenosis or mechanical heart valves. This subgroup comprised patients with AF not attributable to valvular abnormalities and accounted for the majority of cases in the study population. The inclusion criteria of the study were as follows: i) patients had AF, regardless of whether they presented with the first episode or recurrent episodes of AF., ii)

    Comments 7 : Authors concluded that 2.5 mg fixed dose may be helpful In patients with major bleeding, however, table 3 showed no difference in bleeding outcomes, the conclusion is not fully supported by data.

Response 7: We agreed with the reviewer’s comment and modified the conclusion of the study.

Abstract

Conclusion: In Thai patients with AF, warfarin starting dose may be use the 2.5-mg dose,3-mg dose or simplified warfarin dosing formula Whereas 5-mg dose should be avoided due to the high risk of overdosing. A prospective study evaluating different warfarin starting strategies should be pursued. 

We have modified the Conclusion section to emphasize this point as below, Page 7, line 284

Conclusions

Our study suggests that in Thai patients with AF, warfarin starting dose may be use the 2.5-mg dose,3-mg dose or simplified warfarin dosing formula Whereas 5-mg dose should be avoided due to the high risk of overdosing. A prospective study evaluating different warfarin starting strategies should be pursued.

We change the result in the Bleeding outcomes part from Table 3 to Supplementary Table A1(seen in Appendix) as below, Page 6, line 213

Bleeding Outcomes

   Major bleeding was found in 7.75% of patients. Intracranial hemorrhage was found in 4.54% and gastrointestinal bleeding was found in 3.78%. We compared the major bleeding, intracranial bleeding, and GI bleeding with three groups of initiating warfarin dose and there was no statistical significance (Supplementary Table A1)

Table A1. Bleeding events in warfarin using patients classified by actual warfarin starting dose

Actual warfarin starting dose

1-1.5 mg

2-2.5 mg

3 mg

p-value

Major bleeding

7(4.07%)

4(7.27%)

8(9.88%)

0.18742

Intracranial hemorrhage

4(2.33%)

2(3.64%)

5(6.17%)

0.30612

GI bleeding

3(1.74%)

2(3.64%)

5(6.17%)

0.17636

    Comments 8: Authors reported “In the present study, we compared the performance of a simplified warfarin dosing 228 formula, fixed 2.5-mg dose, fixed 3-mg dose, and fixed 5-mg dose strategy in predicting 229 the optimal maintenance dose of warfarin”, however, data in the study were on 1-1.5, 2.5 or 3 mg, why?

Response 8:  The numbers 1-1.5 mg, 2-2.5 mg and 3 mg in Table 2 and Table A1 (Appendix) were actual warfarin starting dose. However, the optimal maintenance dose of warfarin was defined as the warfarin dose that resulted in an INR 2.0 to 3.0 for at least two consecutive follow-ups after the warfarin initiation.

Finally, the simplified warfarin dosing formula, fixed 2.5-mg dose, fixed 3-mg dose and fixed-5 mg dose were different strategies of warfarin starting dose.

In order to avoid confusion, we modified the tables.

Table 2 Median of days to achieve INR target in each actual warfarin starting dose

Actual warfarin starting dose

Days (Median, IQR)

1-1.5 mg

101 (45, 203)

2-2.5 mg

81 (16,184)

3 mg

77 (34, 213)

Table A1. Bleeding events in warfarin using patients classified by actual warfarin starting dose

Actual warfarin starting dose

1-1.5 mg

2-2.5 mg

3 mg

p-value

Major bleeding

7(4.07%)

4(7.27%)

8(9.88%)

0.18742

Intracranial hemorrhage

4(2.33%)

2(3.64%)

5(6.17%)

0.30612

GI bleeding

3(1.74%)

2(3.64%)

5(6.17%)

0.17636

Comments 9: The endpoint of the study is not well describe, I suggest to rephrase the methods section.

Response 9:  We acknowledge the need for clarity regarding the study endpoint. To address this, we propose rephrasing the methods section to provide a more detailed description of the study endpoint.  We rephrase the methods section as described before.

page 2, line 83

Materials and Methods

   The present study was a single-center, retrospective cohort study of patients who were enrolled between 1 October 2011 and 30 September 2021 at a referral hospital. The study included patients with atrial fibrillation (AF), categorized into two main types: non-valvular AF and rheumatic mitral stenosis-associated AF. Non-valvular AF refers to AF occurring in the absence of significant valvular disease, such as mitral valve stenosis or mechanical heart valves. This subgroup comprised patients with AF not attributable to valvular abnormalities and accounted for the majority of cases in the study population. The inclusion criteria of the study were as follows: i) patients had AF, regardless of whether they presented with the first episode or recurrent episodes of AF., ii) patients received continuous warfarin therapy with the target INR of 2.0 to 3.0 at the Outpatient Clinic for more than three months, regardless of their initial OAC treatment status, including the potential use of LMWH.  and iii) patients had achieved INR targets for at least two consecutive follow-up visits. Sequential exclusion criteria were applied to mitigate confounding factors and ensure the homogeneity of the study cohort. These criteria included the exclusion of patients with mechanical heart valves, post-mitral valve surgery, venous thromboembolism (VTE), foreign patients, and those failing to achieve therapeutic international normalized ratio (INR) levels (2.0 to 3.0) for at least two consecutive follow-up visits. The electronic medical records of patients were reviewed. Baseline characteristics, including age, body weight, body height, comorbidities, concurrent medications, actual warfarin dose, and INR levels were collected. The study protocol was approved by the institutional Medical Ethics Committee project approval number S002b/65ExPD.

   In our country, warfarin is typically marketed in tablet form, with common dosages including 2 mg, 3 mg, and 5 mg tablets. When prescribing warfarin, dosages such as 1.5 mg, 2 mg, 2.5 mg, 3 mg, or 5 mg may be required based on factors such as patient character-istics and target INR range. To achieve these dosages, healthcare providers may prescribe a combination of different tablet strengths or tablet splitting that can achieve the desired dosage.

   Warfarin dosage was prescribed and adjusted by treating physicians based on INR results. We collected the dose of warfarin when INR achieved the target range of 2.0 to 3.0 in two consecutive follow-ups and observed the duration required to achieve the INR tar-get in the patients for whom we have data regarding the initiation of warfarin.

   The warfarin dosing formula used in the present study was derived using the algo-rithm described by Pongbangli N et al. and Sarapakidi et al.(13) The simplified warfarin dosing formula (mg/day) was = 3.2 – (0.03 × age (years)) + (0.02 × body weight (kg)) (10% dose reduction if the presence of heart failure or stroke) - Warfarin dosing Formula 1. (13, 14).

We adapted a simplified formula by omitting a history of heart failure and stroke: (mg/day) was = 3.2 – (0.03 × age (years)) + (0.02 × body weight (kg)) without using body weight and the presence of heart failure or stroke - Warfarin dosing Formula 2.

   The other dose strategies commonly used in practice were 2.5 mg-dose, 3 mg-dose, and 5 mg-dose.

   The actual warfarin dose was defined as the warfarin dose that resulted in an INR 2.0 to 3.0 for at least two consecutive follow-ups after the warfarin initiation. The optimal dos-age was defined as the difference from the actual dose beginning within 20%.

   The simplified formula was developed at Chiang Mai, Thailand, and this study sought external validation at Sakaeo province, Thailand. This two-site validation ensures the formula's applicability and reliability across different geographic regions and patient populations. Additionally, we compared the actual dose of warfarin administered to pa-tients with the warfarin dose strategies. These strategies included doses of 2.5 mg, 3 mg, and 5 mg, this comparison aimed to assess the appropriateness and effectiveness of these dose strategies in achieving the target INR range of 2.0 to 3.0.

   The primary endpoint of the study is to compare the proportion of patients falling into appropriate dosing, underdosing, and overdosing when using the simplified warfarin dosing formula, with doses of 2.5 mg, 3 mg, and 5 mg. The secondary endpoints include the actual warfarin starting dose, days to achieve target INR, and bleeding outcomes.

4. Response to Comments on the Quality of English Language

Point 1: I am not qualified to assess the quality of English in this paper

Response: Thank you for your acknowledgment. If you have any other specific areas you'd like to address or if there are any further concerns regarding the paper, please feel free to let me know.

5. Additional clarifications

none

Reviewer 2 Report

Comments and Suggestions for Authors

Review comments on Manuscript entitled “Optimal Strategies to Select Warfarin Dose for Asian Patients with Atrial Fibrillation.”

Authors: Anunya Ujjin, Wanwarang Wongcharoen, Arisara Suwanagool, Chatree Chaiadisaksopha*

In this paper entitled “Optimal Strategies to Select Warfarin Dose for Asian Patients with Atrial Fibrillation”.The paper is aiming to investigate the most optimal warfarin dosing plans for preventing stroke and systemic embolism for patients who suffer the atrial fibrillation (AF) in Thailand. The topic is interesting. However, the entire organization of the manuscript can be further improved. It looks like the authors haven’t put sufficient efforts in summarizing, analyzing and extracting information as well as formatting. The key takeaways are also buried in the manuscript, especially without good summative visuals. Below are some specific comments for your consideration.

Major comments
1) [Title and standpoint of this paper] Line 2: Since the objective is to investigate the most optimal warfarin dosing strategies for preventing stroke and systemic embolism in Thai patients with atrial fibrillation, I suppose the title could be more specific to the patients in Thailand instead of Asian patients. However, throughout the paper, the author stated that the investigation covering the Asian patients, which is not convincible due to the limited cohort. If you would like to keep the same title and standpoint, you probably need to increase the cohort to cover various races in the Asian
countries, etc.

2) [Abstract] The abstract is a good place to highlight the main methodology and findings of the study. For instance, you may want to specify your methods, which is not detailed and clear.

3) [Conclusion] Similar to abstracts, the conclusion is also a place to highlight the main takeaway from this study. But I did not see that in the conclusion. You may want to modify. And any future recommendations/ suggestions/ limitations for this study?

Specific comments
1) [Font color] Line 20-28 & Line 147-160: Please keep consistent of the font color – dark grey to black.

2) [Font size] Line 12-32 & Line 202-217…: I am not sure if the journal  requires a different font size in the Abstract part than that in the main body since I was provided with PDF manuscript. But, if not, please keep consistent of the font size.

3) [Indentation] Line 98: keep consistent of the indentation for each paragraph.

4) [Figure 1] Line 106-109: Improvement the layout of the flow chart accordingly since it is not straightforward to extract the information that the author would like to deliver.

5) [Table 1] Line 118-121: The title of table should be centered, and the improvement of the table quality and style to be clearer (with/without boundary lines such as Three-line form.) instead of a screenshot.

6) [Figure 2] Line 135: Center the Figure title.

7) [Figure 3] Line 142-143: Center both the Figure and Figure title.

8) [Table 3] Line 166: Center the Table title.

9) [Figure 4] Line 167-168: Center the Figure.

10) [Reference] The style, formatting and font size, etc. of the section need to be improved.

11) [Symbol] Line 242: “.” to “,” In Asian patients with AF, and “T” to the fixed…

12) Overall, the formatting of the paper needs to be improved and revised a lot to fulfill the publication standards. Please double check the requirements of formatting from the Journal and adjust them accordingly.

Author Response

Response to Reviewer 2 Comments

  1. Summary

Thank you very much for taking the time to review this manuscript. Please find the detailed

responses below and the corresponding revisions in track changes in the re-submitted files.

2. Questions for General Evaluation

Reviewer’s Evaluation

Response and Revisions

Does the introduction provide sufficient background and include all relevant references?

Must be improved

We have made revisions to the new version as outlined below

Are all the cited references relevant to the research?

Yes

We have made revisions to the new version as outlined below

Is the research design appropriate?

Must be improved

We have made revisions to the new version as outlined below

Are the methods adequately described?

Must be improved

We have made revisions to the new version as outlined below

Are the results clearly presented?

Must be improved

We have made revisions to the new version as outlined below

Are the conclusions supported by the results?

Can be improved

We have made revisions to the new version as outlined below

  1. Point-by-point response to Comments and Suggestions for Authors

Comments 1:

In this paper entitled “Optimal Strategies to Select Warfarin Dose for Asian Patients with Atrial Fibrillation”.The paper is aiming to investigate the most optimal warfarin dosing plans for preventing stroke and systemic embolism for patients who suffer the atrial fibrillation (AF) in Thailand. The topic is interesting. However, the entire organization of the manuscript can be further improved. It looks like the authors haven’t put sufficient efforts in summarizing, analyzing and extracting information as well as formatting. The key takeaways are also buried in the manuscript, especially without good summative visuals. Below are some specific comments for your consideration.

Major comments
1) [Title and standpoint of this paper] Line 2: Since the objective is to investigate the most optimal warfarin dosing strategies for preventing stroke and systemic embolism in Thai patients with atrial fibrillation, I suppose the title could be more specific to the patients in Thailand instead of Asian patients. However, throughout the paper, the author stated that the investigation covering the Asian patients, which is not convincible due to the limited cohort. If you would like to keep the same title and standpoint, you probably need to increase the cohort to cover various races in the Asian
countries, etc.

Response:

Thank you for your valuable feedback regarding the title and standpoint of the paper. The objective of our study is indeed to investigate the most optimal warfarin dosing strategies for preventing stroke and systemic embolism specifically in Thai patients with atrial fibrillation (AF). We understand your concern about the discrepancy between the title and the focus on Thai patients. Therefor, I have modified the title as below, line 2

Optimal Strategies to Select Warfarin Dose for Thai Patients with Atrial Fibrillation

Comments 2:

2) [Abstract] The abstract is a good place to highlight the main methodology and findings of the study. For instance, you may want to specify your methods, which is not detailed and clear.

Response: Thank you for your feedback regarding the abstract. We have modified the abstract as suggested.

Modifications:

Background: Warfarin has been the mainstay treatment for the prevention of stroke and systemic thromboembolism in patients with atrial fibrillation (AF). The optimal starting dose of warfarin remains unclear. Objective: To investigate the most optimal dosing strategies for choosing warfarin starting dose in Thai patients with AF. Material and Methods: We enrolled consecutive AF patients who were on warfarin resulting in a target INR of 2.0-3.0. We collected the dose of warfarin when INR achieved the target range of 2.0 to 3.0. The optimal dosage was defined as the difference from the actual dose within 20%. We compared strategies of warfarin dosing, including warfarin dosing formula, 2.5-mg, - mg and 5-mg doses. The primary endpoints were the proportions of patients in appropriate, underdosing, and overdosing categories. Results: Among 1207 patients visiting the Outpatient Clinic between October 2011 and September 2021, 531 patients were identified with AF and INR in the therapeutic range of 2.0-3.0 in at least 2 consecutive visits. The mean age of participants was 68 ± 11 years, and men accounted for 44.4% of the population. The warfarin dosing formula resulted in optimal dosing in 37% and overdosing in 24% of cases, whereas the 2.5-mg, 3-mg dose and 5-mg dose resulted in optimal dosing in 36%, 39%, and 11%, and overdosing in 33%, 44% and 88% of patients, respectively (p < 0.01). Conclusion: In Thai patients with AF, the warfarin starting dose may be used the 2.5-mg dose,3-mg dose or simplified warfarin dosing formula Whereas the 5-mg dose should be avoided due to the high risk of overdosing.  

Comments 3:

3) [Conclusion] Similar to abstracts, the conclusion is also a place to highlight the main takeaway from this study. But I did not see that in the conclusion. You may want to modify. And any future recommendations/ suggestions/ limitations for this study?

Response: We thank the reviewer for the comments. We have modified the conclusion as suggested. Moreover, the future recommendation and limitations were added in the discussion section.

Modifications:

The present study had some limitations, First, the simplified warfarin dosing formula cannot be used in patients who require a target INR of more than 2.0 to 3.0. Second, our study did not include patients with a mechanical heart valve, which may have a different response to the warfarin dose. Third, we did not collect the thrombotic events, which might be complicated from under warfarin dosing. Fourth, we did not collect the data regarding time in the therapeutic range. Instead, patients with stable INR were defined as those who had achieved INR targets for at least two consecutive follow-up visits. Lastly, we excluded 56% of patients in this cohort due to erratic INR. As mentioned earlier, we had to include patients with stable INR in order to precisely predict warfarin dosage. This might consequently affect the generalizability of the findings.

Conclusions

Our study suggests that in Thai patients with AF, the warfarin starting dose may be used the 2.5-mg dose,3-mg dose or simplified warfarin dosing formula Whereas the 5-mg dose should be avoided due to the high risk of overdosing. A prospective study evaluating different warfarin starting strategies should be pursued.

Specific comments
Comments 1) [Font color] Line 20-28 & Line 147-160: Please keep consistent of the font color – dark grey to black.

Response: Thank you for your observation regarding font color consistency. We have corrected it to black font color.

2) [Font size] Line 12-32 & Line 202-217…: I am not sure if the journal  requires a different font size in the Abstract part than that in the main body since I was provided with PDF manuscript. But, if not, please keep consistent of the font size.

Response: Thank you for noting the difference in font size between the Abstract and the main body. This formatting is following the requirements of the MDPI template for the journal.

3) [Indentation] Line 98: keep consistent of the indentation for each paragraph.

Response: Thank you for your observation regarding the indentation for each paragraph. We have corrected it to be consistent with the indentation.

4) [Figure 1] Line 106-109: Improvement the layout of the flow chart accordingly since it is not straightforward to extract the information that the author would like to deliver.

 Response: Thank you for your observation regarding the layout of the flow chart. We have modified it to a new one.

5) [Table 1] Line 118-121: The title of table should be centered, and the improvement of the table quality and style to be clearer (with/without boundary lines such as Three-line form.) instead of a screenshot.

Response: Thank you for your feedback regarding Table 1. We will ensure that the title of the table is centered, and we will make improvements to the table quality and style to enhance clarity.

6) [Figure 2] Line 135: Center the Figure title.

7) [Figure 3] Line 142-143: Center both the Figure and Figure title.

8) [Table 3] Line 166: Center the Table title.

9) [Figure 4] Line 167-168: Center the Figure.

10) [Reference] The style, formatting and font size, etc. of the section need to be improved.

11) [Symbol] Line 242: “.” to “,” In Asian patients with AF, and “T” to the fixed…

Response 6) to 11)  Thank you for your observation regarding the center of the Figure and Figure title , Table, and Table title. We corrected them in all Figures and tables.

12) Overall, the formatting of the paper needs to be improved and revised a lot to fulfill the publication standards. Please double check the requirements of formatting from the Journal and adjust them accordingly.

Response 1: Thank you for your feedback regarding the formatting of the paper. We will thoroughly review the publication standards and requirements of the Journal to ensure that the formatting is improved and revised to meet these standards. We appreciate your suggestions and will make the necessary adjustments to enhance the quality of the manuscript.

  1. Response to Comments on the Quality of English Language

Point 1: English language fine. No issues detected

Response 1: Thank you for your feedback. I'm glad to hear that you found no issues with the English language in the manuscript. If you have any further comments or questions regarding the content or structure of the manuscript, please feel free to let me know, and I'll be happy to address them.

  1. Additional clarifications

none

Reviewer 3 Report

Comments and Suggestions for Authors

Warfarin (antivitamin K anticoagulant) is an oral anticoagulant with a wild usage/recommendation in countries with a medium or low income.

But note that it is not highly effective! So please correct line 54.

For the general part - please extend this part highlighting the pros and cons of warfarin usage. Note the dietary limitations for patients who are using this type of drug. Present the benefits of prescribing this drug. Analyze the data for immediate and long-term benefits. Note the occurrence of stroke among patients treated with antivitamine K and DOACs. 

You may evaluate the following sources: Kundnani et al, selecting the right anticoagulant for stroke prevention in atrial fibrillation; Lip G et al, Increased markers of thrombogenesis in chronic atrial fibrillation: effects of warfarin treatment; Kreutz R et al, Risk profiles and treatment patterns in atrial fibrillation patients with chronic kidney disease receiving or not receiving anticoagulation therapy

For materials and methods - please add the graphical pathway of inclusion/exclusion criteria in this section. Mention why you excluded amiodarone-treated patients because amiodarone-anticoagulant therapy is frequently used in atrial fibrillation patients if those patients are not considered with permanent atrial fibrillation. Are those patients at the first episode of atrial fibrillation, is this atrial fibrillation a recurrent arrhythmia. Were they treated from the beginning with OAC? If not provide data from the length of the no-anticoagulant therapy period. Introduce the purpose of this study (end-point). Was the aim of this study only the correlation of the dose administrated and the INR value? If yes this is not really important. The INR values were constant in the investigated period? What is the time spent outside of the therapeutic range? Have you investigated this fact? What was the time until the first bleeding? If the bleeding was at the beginning did this patient use the overlap with heparin or LWH? What was the period from the beginning of treatment until the achievement of a therapeutic range of INR? Are these patients using anti-inflammatory drugs (such as diclofenac or others)? 

In the results - please add information about the stroke occurrence, the rate and the length of hemoragical transformation of ischemic stroke. 

Round 2

Reviewer 1 Report

Comments and Suggestions for Authors

I have no further comments

Reviewer 2 Report

Comments and Suggestions for Authors

N/A